# Calcium sulfate beads made with antibacterial essential oil-water emulsions exhibit growth inhibition against *Staphylococcus aureus* in agar pour plates

**Allison N. Hawkins**[ID], **Sara J. Licea**[°], **Sierra A. Sleeper**[°], **Matthew C. Swearingen**[ID]*

Department of Biological Sciences, Florida Gulf Coast University, Fort Myers, Florida, United States of America

° These authors contributed equally to this work.
* swearingen.45@gmail.com

**Data Availability Statement:** All relevant data are within the paper. There are no supporting files.

## Abstract

Calcium sulfate bone void filler beads are fully absorbable in the body, and are often used in complicated orthopedic infection cases to release a relatively high dose of antibiotics locally to the body site over time. However, the antibiotic resistance crisis and/or inability to treat chronic biofilm infections remains to be a formidable and increasing health threat. In this report, we tested the hypothesis that plant essential oils (PEOs) with anti-staphylococcal qualities could inhibit the growth of *Staphylococcus aureus* (a major etiological agent of peri-prosthetic joint infection) in agar pour plates when infused in calcium sulfate beads. To begin, we conducted a screen of 57 single plant PEOs for anti-staphylococcal activity via disk diffusions assays. We observed that 55/57 of the PEOs had significant growth inhibitory activity compared to the null hypothesis, and 41/57 PEOs exhibited activity similar-to-or-higher-than a vancomycin minimum inhibitory control. When PEOs were infused in beads, we observed that 17/57 PEOs tested exhibited significant bacterial growth inhibition when encased in *S. aureus*-seeded agar compared to a null hypothesis of six millimeters (bead size). However, none of the PEO-beads had activity similar to a vancomycin bead control made according to a clinically relevant formula. To the best of our knowledge, this is the first report and screen of PEOs for growth inhibitory activity when infused in lab-made calcium sulfate beads. These data indicate that antibacterial PEOs warrant further investigations, and may be useful in developing new treatment strategies for periprosthetic joint infection.

## Introduction

*Staphylococcus aureus* is a main causative agent of bone and joint infections in orthopedic medicine [1], and is a formidable Gram positive pathogen. *S. aureus* is also responsible for numerous other types of bacterial diseases including skin and soft tissue infections [2], sepsis [3], and respiratory illnesses [4]. In addition to the arsenal of virulence factors that *S. aureus* produces during infection [5], *S. aureus* also forms biofilms that resist both host defenses and

Table 1 of the manuscript lists all the data in its entirety.

**Funding:** This work was supported with departmental yearly budget funds from the Biological Sciences Department at Florida Gulf Coast University. In other words, supplies were purchased with the same departmental funds that would have been used to order laboratory supplies for teaching laboratories. The funders played no role in study design, data collection and analysis or the preparation of the manuscript. The funders permitted the preparation of the manuscript for publishing. No authors received salary from the funders or with the funds supplied from the biological sciences department.

**Competing interests:** The authors have declared that no competing interests exist.

chemical/antibiotic attack [5, 6]. What is more, a large portion of *S. aureus* infections are difficult to treat because they are caused by methicillin-resistant *S. aureus* (MRSA) [7], whereas, methicillin is a first line of treatment for *S. aureus* infections [7]. For MRSA infections, vancomycin is often used as the next line of treatment [8], however, vancomycin-resistant *S. aureus* (aka VRSA) isolates have also emerged in which case daptomycin may be implemented [9].

In orthopedics, the use of vancomycin-loaded fully resorbable therapeutic-grade calcium sulfate ($CaSO_4$) bone void filler beads (beads), and/or non-absorbable poly(methylmethacrylate) (PMMA) is a common tactic for intractable enterococci infections, including MRSA [10]. In infection cases where an orthopedist elects to use antibiotic-loaded $CaSO_4$ beads, pharmaceutical-grade antibiotic powders are hand-blended with purified $CaSO_4$ powder, which is then mixed with a proprietary water-based solvent (which accelerates bead-hardening). The resulting paste is spackled into an elastomer bead mold (the beads solidify in minutes), and they are then expressed out of the mold for use in the body site.

The mixing of antibiotics into $CaSO_4$ beads (or PMMA) occurs primarily via "off-label" use in U.S. operatories, and at the discretion of the surgeon [10, 11]. The threat of antibiotic resistance in infection cases however, especially in orthopedics, is a modern critical concern [10, 12], and the discovery and/or development of new or novel antimicrobials is crucial to treating modern and future infection cases. One potentially good, abundant, and affordable source of natural antimicrobials are plant distillates (aka plant essential oils, PEOs). Indeed, a wealth of literature exists reporting and reviewing the antimicrobial efficacy of PEOs against important microbes, including *S. aureus*, as well as the suspected mechanism(s) by which PEOs elicit antimicrobial effects [13].

Existing reports have modeled and researched the antimicrobial qualities and kinetics of lab-made antibiotic-loaded $CaSO_4$ beads against both Gram positive and Gram negative orthopedic pathogens, including S. aureus [10, 14–16]. In this report, however, we tested the hypothesis that PEOs with anti-staphylococcal properties would elicit growth inhibition against *S. aureus in vitro* when infused in $CaSO_4$ beads that mimic those made for use in orthopedic infection cases. To test this, we loaded PEOs into lab-made $CaSO_4$ beads and observed if PEOs could elicit *S. aureus* growth inhibition in agar pour plates. In addition to identifying a number of PEOs that elicit anti-staphylococcal activity, we also note that initial disk diffusion assays were not reliably predictive of which PEOs would also elicit growth inhibition when infused in beads.

## Materials and methods

### Strains, media, growth conditions, and plant essential oils tested

This study used *S. aureus* ATCC® 25923™ (Rosenbauch) as model (orthopedic) pathogenic organism. For general culturing purposes, *S. aureus* was grown and maintained using BD Bacto™ tryptic soy broth (TSB) or TSB-agar (Fisher Scientific, Hampton, NH). Prior to performing disk diffusion assays, bacterial strains were grown to stationary phase overnight in TSB with shaking aeration at 37˚C. For susceptibility testing, BD™ BBL™ MHII agar (Fisher Scientific) was prepared fresh according to the manufacturer's instructions and dried at ambient room temperature for 24 hours. All susceptibility tests (see below) were carried out at 37˚C in a humidified chamber with 5% $CO_2$ atmospheric buffering. PEOs tested include Arborvitae, Balsam Fir, Basil, Bergamot, Black Pepper, Black Spruce, Blue Tansy, Cardamom, Cassia, Cedarwood, Roman Chamomile, Cilantro, Cinnamon Bark, Clary Sage, Clove, Copaiba, Coriander, Cypress, Douglass Fir, Eucalyptus, Fennel, Frankincense, Geranium, Ginger, Grapefruit, Green Mandarin, Helichrysum, Jasmine, Juniper Berry, Lavendar, Lemon, Lemongrass, Lime, Marjoram, Melaleuca, Melissa, Myrrh, Neroli, Oregano, Patchouli, Peppermint,

Petitgrain, Pink Pepper, Rose, Rosemary, Sandalwood, Hawaiian Sandalwood, Siberian Fir, Spearmint, Spikenard, Tangerine, Thyme, Turmeric, Vetiver, Wild Orange, Wintergreen, and Ylang Ylang.

## Determining the growth inhibitory activity of 57 essential oils via disk diffusion assays

Stationary phase cultures of *S. aureus* were subcultured 1:100 in fresh sterile TSB and mixed thoroughly. Then, 100 μL of the *S. aureus* sub-culture was spread onto BD BBL™ MHII agar plates (100 mm diameter) for a final cell volume equivalent to 1/1000th of the original culture density, or approximately 5 x $10^5$ CFU per plate. Next, six mm blank sterile paper filter disks (Fisher Scientific) were aseptically transferred to each spread plate, and 20 μL of each PEO was applied to their respective disks. PEOs observed to have high potency (having full or relatively large zones of inhibition) were tested on plates in the absence of other PEOs so that the zones of inhibition (ZOIs) could be appreciated. All plates were incubated as described above (see strains, media, and growth conditions) for approximately 24 hours ("overnight"). After incubation, the ZOIs on the plates were measured with a Fisher brand metric ruler. The student's T test was used to determine significance, see Table 1.

## Determining the pour plate ZOI of PEO- and vancomycin-loaded beads

For PEO-bead growth inhibition assays, we first created 1:1 vortex emulsifications of PEOs in sterile $dH_2O$ in 50 mL sterile falcon tubes. Emulsifications totaled 4 mL in volume. PEO:water mixtures were emulsified by vortex at full speed for 15 seconds, and then allowed to rest and separate for 5 minutes. Then, a second vortex emulsification was repeated, the PEO:water emulsates were promptly mixed into 5 g of reagent-grade $CaSO_4$ (Sigma-Aldrich, St. Louis, MO, USA) in an alcohol-sanitized plastic weigh boat, and then hand-mixed using a sanitized spatula (Biocomposites, UK). The PEO-loaded $CaSO_4$ paste was molded into six mm sized hemispherical beads using an estastomer bead mold (Biocomposites, UK). The $CaSO_4$ beads were allowed to harden for one hour, and were then expressed from the mold into a sterile petri plate. Vancomycin-loaded $CaSO_4$ paste was prepared according to a clinically relevant formula of 1 g of vancomycin HCL (Fisher Scientific) per 10 cc of $CaSO_4$, and hardened beads were prepared similarly to PEO-beads described above. To test the beads for growth inhibitory activity, fresh sterile molten MHII agar was cooled to 43˚C in a water bath. Then, an overnight grown culture of *S. aureus* was subcultured 1:1000 (approximately 5 x $10^5$ CFU) in the molten media and stirred thoroughly. The seeded MHII agar was quickly poured into empty sterile petri plates containing pre-positioned PEO or vancomycin beads and allowed to solidify. Hardened pour plates were incubated overnight as described above, and the diameter of the resulting pour plate ZOI (PPZOI) was measured, if applicable. The student's T test was used to determine significance, see Table 1.

## Results

### Growth inhibitory activity of 57 PEOs against planktonic *S. aureus*

Initially we wanted to test the anti-staphylococcal activity of our PEO collection via disk diffusion assays. All of the PEOs tested demonstrated significant growth inhibitory activity against planktonic *S. aureus* (Table 1) when comparing ZOIs to a null hypothesis of 6 mm (disk size and LOD), except for neroli and wintergreen. 19/57 PEOs tested, including arborvitae, balsam fir, basil, cardamom, cassia, cilantro, cinnamon bark, clary sage, cypress, Douglas fir, frankincense, lemongrass, melaleuca, Melissa, oregano, peppermint, Siberian fir, tangerine, and

**Table 1. Average growth inhibition ZOIs and one sample T tests for all PEOs tested in disk diffusion (DDA) and pour plate bead assays.**

| PEO | Scientific Name | DDA ZOI (mm) | SEM (± mm) | DDA ZOI vs. NH#1 (p values) | Activity* Relative to Vancomycin ZOI | Bead PPZOI (mm) | SEM (± mm) | T-test vs. NH#2 (p values) | PPZOI vs. Vancomycin Bead (p values) |
|---|---|---|---|---|---|---|---|---|---|
| Arborvitae | *Thuja plicata* | 28.5 | 1.1 | <0.0001 | Higher | 17.4 | 0.8 | <0.0001 | Lower |
| Balsam Fir | *Abies balsamea* | 37.3 | 1.5 | <0.0001 | Higher | 19.2 | 0.3 | <0.0001 | Lower |
| Basil | *Ocimum basilicum* | 33.7 | 4.7 | 0.002 | Higher | - | N/A | N/A | Lower |
| Bergamot | *Citrus bergamia* | 17.6 | 3.6 | 0.0225 | Similar | 5.6 | 1.2 | 0.7675 | Lower |
| Black Pepper | *Citrus bergamia* | 18.9 | 1.3 | 0.0002 | Similar | 2.1 | 1.1 | 0.0044 | Lower |
| Black spruce | *Picea mariana* | 33.0 | 5.2 | 0.0036 | Similar | 14.7 | 1.0 | 0.0002 | Lower |
| Blue Tansy | *Tanacetum annuum* | 15.3 | 1.4 | 0.0012 | Similar | - | N/A | N/A | Lower |
| Cardamom | *Elettaria cardamomum* | 25.3 | 1.0 | <0.0001 | Higher | 5.7 | 3.0 | 0.9307 | Lower |
| Cassia | *Elettaria cardamomum* | 32.2 | 1.7 | <0.0001 | Higher | 21.8 | 0.6 | <0.0001 | Lower |
| Cedarwood | *Elettaria cardamomum* | 12.7 | 0.7 | 0.0002 | Lower | 2.6 | 1.1 | 0.0103 | Lower |
| Rom. Chamomile | *Elettaria cardamomum* | 18.3 | 1.6 | 0.0005 | Similar | - | N/A | N/A | Lower |
| Cilantro | *Coridandrum sativum* | 80.0 | 0.0 | <0.0001** | Higher | 7.4 | 1.2 | 0.244 | Lower |
| Cinnamon Bark | *Cinnamomum zeylanicum* | 29.8 | 2.2 | 0.0001 | Higher | 20.7 | 1.5 | <0.0001 | Lower |
| Clary Sage | *Salvia sclarea* | 21.0 | 1.6 | 0.0003 | Higher | 8.0 | 1.9 | 0.3126 | Lower |
| Clove | *Eugenia carophyllata* | 16.0 | 0.9 | <0.0001 | Similar | 7.1 | 1.5 | 0.5041 | Lower |
| Copaiba | *Copaifera* | 16.6 | 1.5 | 0.0008 | Similar | 0.5 | 0.5 | <0.0001 | Lower |
| Coriander | *Coriandrum sativum* | 21.7 | 3.9 | 0.0101 | Similar | 5.9 | 2.2 | 0.9564 | Lower |
| Cypress | *Cupressus sempervirens* | 32.0 | 5.3 | 0.0043 | Higher | 12.0 | 2.2 | 0.0218 | Lower |
| Douglas Fir | *Pseudotsuga menziessii* | 29.7 | 2.2 | 0.0001 | Higher | 12.2 | 0.8 | <0.0001 | Lower |
| Eucalyptus | *Eucalyptus radiata* | 22.8 | 3.8 | 0.0071 | Similar | 11.9 | 2.6 | 0.0449 | Lower |
| Fennel | *Foeniculum vulgare* | 13.7 | 1.7 | 0.0058 | Similar | 1.8 | 0.9 | 0.0007 | Lower |
| Frankincense | *Boswellia* | 31.7 | 4.9 | 0.0035 | Higher | 10.3 | 1.6 | 0.0175 | Lower |
| Geranium | *Pelargonium graveolens* | 15.8 | 2.0 | 0.0046 | Similar | 7.0 | 2.5 | 0.6917 | Lower |
| Ginger | *Zingiber officinale* | 12.2 | 0.8 | 0.0006 | Lower | 3.7 | 1.3 | 0.0981 | Lower |
| Grapefruit | *Citrus x paradisi* | 14.7 | 3.1 | 0.0389 | Similar | 17.6 | 0.5 | <0.0001 | Lower |
| Green Mandarin | *Citrus reticulata* | 11.8 | 0.4 | <0.0001 | Lower | - | N/A | N/A | Lower |
| Helichrysum | *Helichrysum italicum* | 16.0 | 3.2 | 0.0253 | Similar | 10.2 | 0.6 | 0.0003 | Lower |
| Jasmine | *Jasminum grandiforum* | 11.2 | 1.3 | 0.0106 | Lower | - | N/A | N/A | Lower |
| Juniper Berry | *Juniperis communis* | 14.0 | 2.3 | 0.0171 | Similar | 9.6 | 1.7 | 0.057 | Lower |
| Lavender | *Lavandula angustifolia* | 22.2 | 2.6 | 0.0016 | Similar | 7.7 | 1.2 | 0.1914 | Lower |
| Lemon | *Citrus limon* | 21.3 | 2.5 | 0.0018 | Similar | 9.9 | 1.4 | 0.0249 | Lower |

(*Continued*)

**Table 1.** (Continued)

| PEO | Scientific Name | DDA ZOI (mm) | SEM (± mm) | DDA ZOI vs. NH#1 (p values) | Activity* Relative to Vancomycin ZOI | Bead PPZOI (mm) | SEM (± mm) | T-test vs. NH#2 (p values) | PPZOI vs. Vancomycin Bead (p values) |
|---|---|---|---|---|---|---|---|---|---|
| Lemongrass | *Cymbopogon flexuosus* | 29.2 | 2.8 | 0.0004 | Higher | 10.1 | 0.5 | <0.0001 | Lower |
| Lime | *Citrus aurantifolia* | 20.7 | 3.3 | 0.0069 | Similar | 7.7 | 1.7 | 0.3305 | Lower |
| Marjoram | *Origanum majorana* | 19.7 | 2.8 | 0.0044 | Similar | 6.1 | 1.9 | 0.9524 | Lower |
| Melaleuca | *Melaleuca alternifolia* | 23.2 | 2.5 | 0.001 | Higher | 10.3 | 2.4 | 0.1045 | Lower |
| Melissa | *Melissa officinalis* | 19.8 | 0.6 | <0.0001 | Higher | 1.9 | 1.0 | 0.002 | Lower |
| Myrrh | *Commiphohra myrrha* | 10.8 | 0.4 | 0.0082 | Lower | 5.8 | 1.2 | 0.8436 | Lower |
| Neroli | *Citrus x aurantium* | 17.8 | 5.1 | 0.0689 | Similar | 2.0 | 0.9 | 0.0008 | Lower |
| Oregano | *Origanum vulgare* | 28.5 | 3.0 | 0.0006 | Higher | 15.3 | 0.7 | <0.0001 | Lower |
| Patchouli | *Pogostemon cablin* | 12.8 | 1.0 | 0.0009 | Lower | 5.9 | 1.3 | 0.9232 | Lower |
| Peppermint | *Mentha piperita* | 20.0 | 1.4 | 0.0002 | Higher | 8.5 | 0.5 | 0.0003 | Lower |
| Petitgrain | *Citrus aurantium* | 22.8 | 3.4 | 0.0045 | Similar | 3.0 | 1.1 | 0.0165 | Lower |
| Pink Pepper | *Schinus terebinthifolia* | 10.0 | 0.3 | <0.0001 | Lower | 2.3 | 1.0 | 0.0027 | Lower |
| Rose | *Rosa damascena* | 8.1 | 0.4 | 0.0026 | Lower | - | N/A | N/A | Lower |
| Rosemary | *Rosmarinus officinalis* | 29.5 | 7.4 | 0.0252 | Similar | 6.8 | 1.7 | 0.6639 | Lower |
| Sandalwood | *Santalum album* | 10.0 | 1.1 | 0.0147 | Lower | 2.2 | 1.1 | 0.0086 | Lower |
| Haw. Sandalwood | *Santalum paniculatum* | 9.3 | 1.0 | 0.0218 | Lower | 3.7 | 1.7 | 0.2339 | Lower |
| Siberian Fir | *Abies sibirica* | 31.2 | 5.4 | 0.0054 | Higher | 13.2 | 0.2 | <0.0001 | Lower |
| Spearmint | *Mentha spicata* | 22.7 | 4.2 | 0.0111 | Similar | - | N/A | N/A | Lower |
| Spikenard | *Nardostachys jatamansi* | 12.7 | 1.1 | 0.0017 | Lower | 4.7 | 2.4 | 0.6349 | Lower |
| Tangerine | *Citrus reticulata* | 29.0 | 4.3 | 0.0031 | Higher | 7.4 | 3.6 | 0.7139 | Lower |
| Thyme | *Thymus vulgaris* | 42.5 | 3.4 | 0.0001 | Higher | 13.1 | 1.6 | 0.0029 | Lower |
| Turmeric | *Curcuma longa* | 8.6 | 0.2 | <0.0001 | Lower | 2.6 | 1.3 | 0.0295 | Lower |
| Vetiver | *Vetiveria zizanioides* | 10.4 | 0.8 | 0.0027 | Lower | - | N/A | N/A | Lower |
| Wild orange | *Citrus sinensis* | 9.3 | 0.8 | 0.0084 | Lower | 1.1 | 1.1 | 0.0034 | Lower |
| Wintergreen | *Gaultheria fragantissima* | 6.7 | 2.3 | 0.7827 | Lower | 1.1 | 1.1 | 0.0023 | Lower |
| Ylang Ylang | *Cananga odorata* | 9.5 | 0.8 | 0.0056 | Lower | - | N/A | N/A | Lower |
| Vancomycin | | 16.0 | 0.3 | <0.0001 | N/A | 24.1 | 0.3 | <0.0001 | N/A |

ZOIs listed represent the average ZOI diameters calculated from three biological replicates, where $n \geq 6$ (up to 12), and variation is represented as the standard error of the means (SEM). P values of $< 0.05$ were considered significant.

* Activity is based on p value comparisons.

** Indicates that cilantro had full inhibition in all DDA trials, therefore we artificially introduced ± 0.5 mm of error to the average value of 80 mm for statistical comparison.

NH#1 = null hypothesis of 6 mm for disk diameter.

NH#2 = null hypothesis of 6 mm for bead diameter.

N/A indicates not applicable.

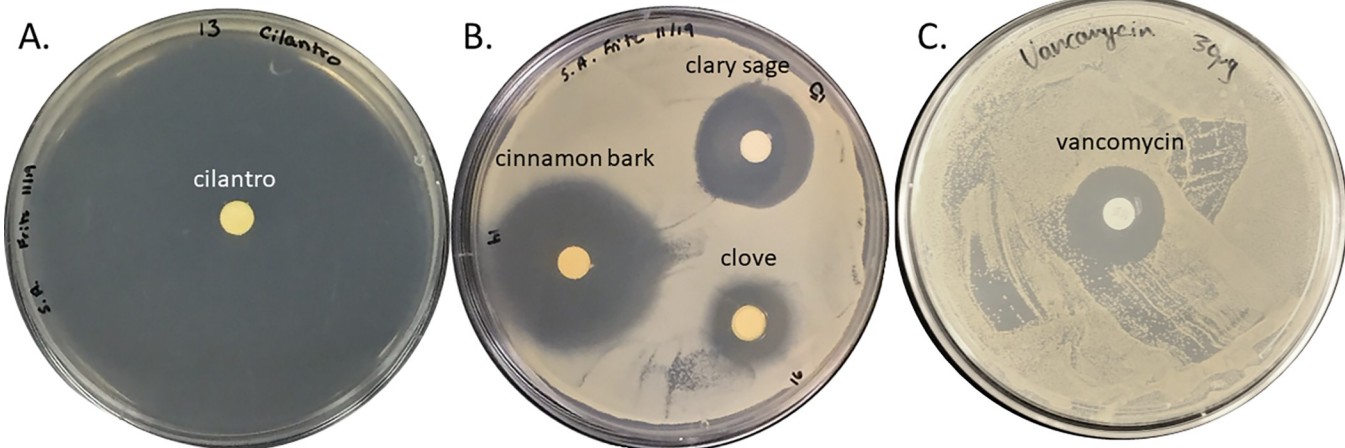

**Fig 1. PEOs inhibit the growth of *S. aureus* in disk diffusion assays.** Representative images of cilantro (A), and cinnamon bark, clary sage, and clove (B) versus a 30 μg vancomycin control (C). Images represent n ≥ 6.

thyme all demonstrated significantly higher ZOIs compared to a vancomycin 30 μg minimum inhibitory concentration control (Fig 1), while an additional 22/57 PEOs tested had ZOIs statistically similar to the vancomycin control (Table 1). Most notably, cilantro PEO exhibited full growth inhibition of *S. aureus* in all replicates (n = 6, Table 1). Cedarwood, ginger, green Mandarin, jasmine, myrrh, patchouli, pink pepper, rose, sandalwood and Hawaiian sandalwood, spikenard, turmeric, vetiver, wild orange, wintergreen, and ylang ylang exhibited ZOIs significantly less than that of the vancomycin control (Table 1).

### Growth inhibitory activity of PEO-beads

Despite a number of the PEOs exhibiting relatively low-or-no activity in disk diffusion assays compared to vancomycin, we opted to screen all 57 PEOs infused in beads against *S. aureus* to better understand how disk diffusion assays might be predictive of PEO-bead activity. We operated under the general hypothesis that the disk diffusion data would positively correlate with PEO-bead growth inhibition. That is, for instance, we hypothesized that PEOs with high anti-staphylococcal activity, such as cilantro, would elicit high growth inhibitory activity when infused in beads.

When comparing PPZOI diameters against a null hypothesis of 6 mm (bead size and LOD), 17/57 PEO-beads including arborvitae, balsam fir, black spruce, cassia, cinnamon bark, cypress, Douglas fir, eucalyptus, geranium, grapefruit, helichrysum, lemon, lemongrass, oregano, peppermint, Siberian fir, and thyme had significant growth inhibitory activity (Table 1 and Fig 2). However, none of the PEOs tested exhibited growth inhibition statistically higher-or-similar-to a vancomycin-bead control (24.1, ± 0.3 mm). Additionally, we observed that a number of PEOs (22/57) which, according to the disk diffusion data, were hypothesized to elicit high growth inhibitory activity when infused in beads failed to do so (Table 1), including basil, bergamot, black pepper, blue tansy, cardamom, cedarwood, cilantro, clary sage, clove, copaiba, coriander, fennel, geranium, juniper berry, lavender, lime, marjoram, melaleuca, Melissa, neroli, petitgrain, rosemary, spearmint, and tangerine.

To conserve materials, we tested multiple PEOs in per plate in some tests (in replicate), and this imposes a potential methodological limit in that there could be imperceivable antagonistic or synergistic effects occurring between proximal PEOs. Tests for multi-PEO synergism or antagonism were within the scope of this research, and would require follow-up testing.

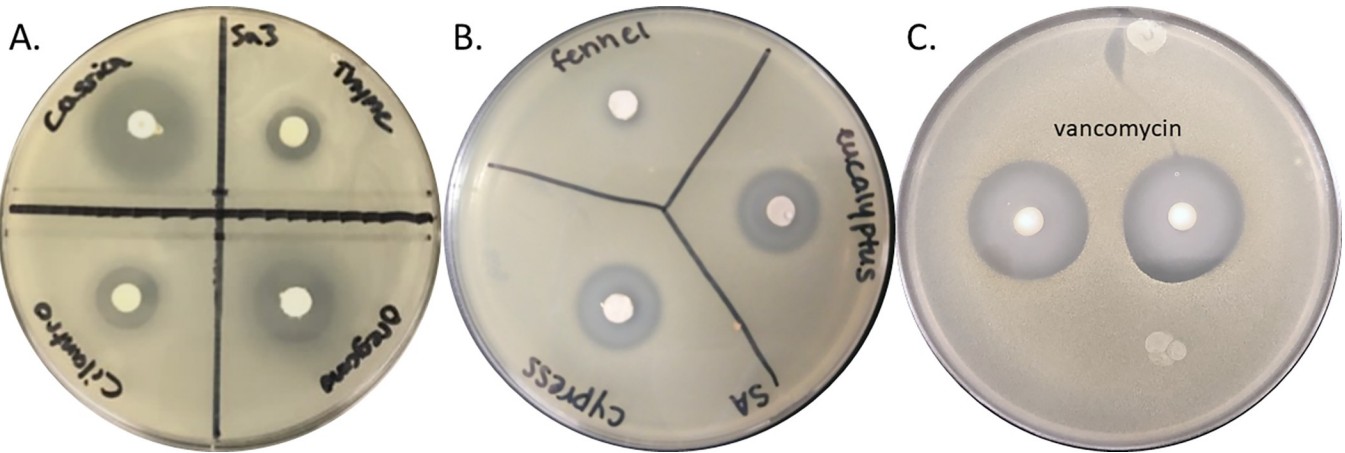

**Fig 2. PEO-loaded beads inhibit *S. aureus* growth in agar pour plates.** Representative images of Cassia, Cilantro, Oregano, and Thyme (A), Fennel, Eucalyptus, and Cypress (B), versus the vancomycin 30 µg control (C, in duplicate) loaded beads inhibiting *S. aureus* growth in three dimensions.

Regardless, and finally, we calculated the hypothetical SOI that would result if the beads were encased in an adequate amount of seeded media (and/or as if placed in a joint space), and statistically compared the SOIs to the vancomycin control. We found that cinnamon bark PEO would elicit a theoretical SOI (5.5 cm$^3$ ±1.1 cm$^3$) statistically similar to vancomycin (7.4 cm$^3$ ± 0.2 cm$^3$).

## Discussion

The antibiotic resistance crisis is well known amongst scientists and physicians. Some creative strategies and inventions for combatting infections, which are gaining considerable focus, include the use of antimicrobial nanoparticles, monoclonal antibodies, antimicrobial peptides, conjugated/tandem antibiotics, vaccines, and phage therapy [17]. Here, we corroborate several decades worth of literature reporting the antimicrobial efficacy of various PEOs against major pathogens (or food spoilage organisms) [13, 18], indicating that PEOs or other plant derivatives represent an abundant and affordable source of natural antimicrobials. Furthermore, we demonstrated that PEOs with anti-staphylococcal properties can be infused in CaSO$_4$ beads and inhibit *S. aureus* growth *in vitro*. We observed several PEOs that were seemingly active against *S. aureus* in agar pour plates, but none that performed as well-or-better than a vancomycin bead control. It is worth considering, though, that the vancomycin beads used in this study were prepared to mimic a clinical formula [19] (see Materials and Methods), which yields beads containing roughly 333 times the minimum inhibitory concentration (30 µg, MIC) for vancomycin against *S. aureus*. With that being said, the average PEO-bead contained roughly 41.4 µL (±2.8 µL) of PEO per bead, which is approximately two times the amount used in disk diffusion assays. Thus, future work remains to be done with regards to optimizing the PEO:water-CaSO$_4$ mixture described here to determine if the volume of PEO can be increase per bead.

Nevertheless, if we statistically analyze PEO-bead PPZOI diameters using a null hypothesis of six mm (bead size and LOD), we find that arborvitae, balsam fir, black spruce, cassia, cinnamon bark, cypress, Douglas fir, eucalyptus, frankincense, grapefruit, helichrysum, lemon, lemongrass, oregano, peppermint, rosemary, sandalwood, spearmint, and thyme have significant growth inhibitory activity (Table 1); all of which were predictable by disk diffusion assays. However, because 22/57 PEOs did not elicit statistically significant growth inhibition when

infused in beads, as we had predicted, we conclude that disk diffusion assays are unreliable at predicting growth inhibitory activity of PEO-beads in agar pour plates. For instance, one major discrepancy we observed was that cilantro bead PPZOIs were almost immeasurable, having an average diameter of 7.4 mm (±1.2 mm), whereas, cilantro PEO completely inhibited *S. aureus* growth in all disk diffusion replicates (Table 1). The discrepancy for cilantro and other PEOs across assays may be related to a reduction in elution rate specific to cilantro PEO when it is infused in $CaSO_4$, and/or the hydrophobicity of the active compound(s) in PEOs when encased in an aqueous medium. Consequently, comparison of PEOs predicted to have bead activity with those that actually did may help us predict which PEOs have antimicrobial hydrosolic compounds capable of eluting into aqueous environments; antibacterial hydrosolic compounds would be attractive compounds to identify and research because of their water solubility.

Encasing beads in bacteria-seeded agar is an interesting way to measure their activity in three dimensions, and might represent a useful *in vitro* approach to measuring the efficacy of potential antimicrobial alternatives, especially for novel compounds that are water soluble. One additional pitfall of the bead pour plate assay is that the normal dissolution process of $CaSO_4$ beads (such that occurs in the weeks following implantation) is not emulated, and this is a crucial aspect of the gradual release of antibiotic locally at a body site. The rate of dissolution is largely related to the fluid levels and movement occurring in the joint space, but typically occurs over a period of several weeks [20]. In a scenario in which antibacterial PEO-beads dissolve over time, we would predict that the PEO loaded into beads would also be gradually released. Thus, although our pour plate assays did not indicate any of the oils to be as effective as a vancomycin bead control, a scenario in which the beads dissolved over time might yield more relevant results. Nevertheless, six of the significantly active PEO-beads had relatively/consistently high activity compared to all others, including arborvitae (17.4 mm), balsam fir (19.2 mm), cassia (21.8 mm), cinnamon bark (20.7 mm), grapefruit (17.6 mm), and oregano (15.28 mm), and these may represent the best candidates for additional study and optimization for use in beads. To the best of our knowledge, this is the first time in which a large set of single plant PEOs have been demonstrated to exhibit growth inhibition of *S. aureus in vitro* when infused in $CaSO_4$ beads.

## Acknowledgments

We would like to thank doTERRA® International for the donation of essential oils in support of this work, especially Dr. David Hill, D.C. and Dr. Cody Beaumont, Ph.D. We would also like to thank Sean Aiken of Biocomposites ® (UK) for his gracious provision of bead making supplies and scientific input.

## Author Contributions

**Conceptualization:** Matthew C. Swearingen.

**Data curation:** Allison N. Hawkins, Sara J. Licea, Sierra A. Sleeper, Matthew C. Swearingen.

**Formal analysis:** Matthew C. Swearingen.

**Funding acquisition:** Matthew C. Swearingen.

**Investigation:** Matthew C. Swearingen.

**Methodology:** Matthew C. Swearingen.

**Project administration:** Matthew C. Swearingen.

**Resources:** Matthew C. Swearingen.

**Software:** Matthew C. Swearingen.

**Supervision:** Matthew C. Swearingen.

**Validation:** Matthew C. Swearingen.

**Visualization:** Matthew C. Swearingen.

**Writing – original draft:** Matthew C. Swearingen.

**Writing – review & editing:** Matthew C. Swearingen.

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
