## [Decision Letter · Decision Letter 0]

14 Jun 2021

PONE-D-21-14096

Calcium sulfate beads made with antibacterial essential oil-water emulsions exhibit growth inhibition against Staphylococcus aureus in agar pour plates.

PLOS ONE

Dear Dr. Swearingen,

Thank you for submitting your manuscript to PLOS ONE. After careful consideration, we feel that it has merit but does not fully meet PLOS ONE’s publication criteria as it currently stands. Therefore, we invite you to submit a revised version of the manuscript that addresses the points raised during the review process.

We look forward to receiving your revised manuscript.

Kind regards,

Arumugam Sundaramanickam, PhD

Academic Editor

PLOS ONE

Additional Editor Comments:

Most of the references are old. Include some recent references between 2019 and 2021.

Journal Requirements:

"This work was supported by intramural funds from the FGCU Department of Biological

Sciences and the FGCU Whitaker Center for Stem Education. We would like to thank doTERRA®

International for the donation of essential oils in support of this work, especially Dr. David Hill,

D.C. and Dr. Cody Beaumont, Ph.D. We would also like to thank Sean Aiken of Biocomposites ®

(UK) for his gracious provision of bead making supplies and scientific input."

Additionally, because some of your funding information pertains to commercial funding, we ask you to provide an updated Competing Interests statement, declaring all sources of commercial funding.

In your Competing Interests statement, please confirm that your commercial funding does not alter your adherence to PLOS ONE Editorial policies and criteria by including the following statement: "This does not alter our adherence to PLOS ONE policies on sharing data and materials.” as detailed online in our guide for authors  http://journals.plos.org/plosone/s/competing-interests.  If this statement is not true and your adherence to PLOS policies on sharing data and materials is altered, please explain how.

Please include the updated Competing Interests Statement and Funding Statement in your cover letter. We will change the online submission form on your behalf.

Reviewers' comments:

Reviewer's Responses to Questions

**Comments to the Author**

1. Is the manuscript technically sound, and do the data support the conclusions?

Reviewer #1: Yes

Reviewer #2: Yes

2. Has the statistical analysis been performed appropriately and rigorously? 

Reviewer #1: Yes

Reviewer #2: Yes

3. Have the authors made all data underlying the findings in their manuscript fully available?

Reviewer #1: Yes

Reviewer #2: Yes

4. Is the manuscript presented in an intelligible fashion and written in standard English?

Reviewer #1: Yes

Reviewer #2: Yes

5. Review Comments to the Author

Reviewer #1: This is a well done study with a clearly defined focus. It shows that anti-microbial oils may have potential for inhibiting S. aureus in calcium sufate beads. The results are also relevant, in addition to the medical aspects, in the food sector.

Minor comment: line 77 and 79 S. aureus in italics.

Reviewer #2: The study is quite impressive and scientifically elaborated. I advise some minor corrections/clarifications:

- Mention the plants under study in the material and methods section.

- Mention scientific names of the plants in Table 1

- Include some representative images of treatments displaying significantly higher ZOIs compared to vancomycin 30 µg control.

-Include representative image of PEO seeded beads

- What was the basis of selecting 30 µg of vancomycin as control?

6. PLOS authors have the option to publish the peer review history of their article (what does this mean?). If published, this will include your full peer review and any attached files.

Reviewer #1: **Yes: **Per Saris

Reviewer #2: No

---

## [Author Response · Author response to Decision Letter 0]

21 Sep 2021

Review Feedback and Comments pasted from decision letter email with our responses in red.

o Complete

o Complete

o Complete

- N/A

- N/A

We look forward to receiving your revised manuscript.

Kind regards,

Arumugam Sundaramanickam, PhD

Academic Editor

PLOS ONE

Additional Editor Comments:

Most of the references are old. Include some recent references between 2019 and 2021.

- Thank you. We updated all the references that could be updated. Some of the references, however, are seminal works or otherwise key pieces of information that support the narrative and project justification. If there are any specific references you would still like us to try to update, please let us know. 

Journal Requirements:

 - We reformatted the references to match the formatting requirements of the journal. 

- Thank you for this comment and the links. This was extremely helpful. We have revised the style of the manuscript exactly as instructed in the links. One issue we were unsure of, though, was table 1, which is quite wide. In order for table 1 to fit underneath the paragraph in which it is first mentioned, we had to flip the pages horizontally. We were not sure if this is appropriate or how we should approach this technical issue. 

"This work was supported by intramural funds from the FGCU Department of Biological

Sciences and the FGCU Whitaker Center for Stem Education. We would like to thank doTERRA®

International for the donation of essential oils in support of this work, especially Dr. David Hill,

D.C. and Dr. Cody Beaumont, Ph.D. We would also like to thank Sean Aiken of Biocomposites ®

(UK) for his gracious provision of bead making supplies and scientific input."

Additionally, because some of your funding information pertains to commercial funding, we ask you to provide an updated Competing Interests statement, declaring all sources of commercial funding.

In your Competing Interests statement, please confirm that your commercial funding does not alter your adherence to PLOS ONE Editorial policies and criteria by including the following statement: "This does not alter our adherence to PLOS ONE policies on sharing data and materials.” as detailed online in our guide for authors http://journals.plos.org/plosone/s/competing-interests. If this statement is not true and your adherence to PLOS policies on sharing data and materials is altered, please explain how.

Please include the updated Competing Interests Statement and Funding Statement in your cover letter. We will change the online submission form on your behalf.

 - Thank you. We have revised the acknowledgements section to read as follows:

“We would like to thank doTERRA® International for the donation of essential oils in support of this work, especially Dr. David Hill, D.C. and Dr. Cody Beaumont, Ph.D. We would also like to thank Sean Aiken of Biocomposites ® (UK) for his gracious provision of bead making supplies and scientific input.”

 - Thank you. We have removed the ‘data not shown” phrase from the manuscript. The manuscript is not significantly altered with/without it. 

Reviewers' comments:

Reviewer's Responses to Questions

Comments to the Author

1. Is the manuscript technically sound, and do the data support the conclusions?

- Thank you!

Reviewer #1: Yes

Reviewer #2: Yes

2. Has the statistical analysis been performed appropriately and rigorously?

Reviewer #1: Yes

Reviewer #2: Yes

- Thank you.

3. Have the authors made all data underlying the findings in their manuscript fully available?

Reviewer #1: Yes

Reviewer #2: Yes

- Thank you.

4. Is the manuscript presented in an intelligible fashion and written in standard English?

Reviewer #1: Yes

Reviewer #2: Yes

- Thank you.

5. Review Comments to the Author

Reviewer #1: This is a well done study with a clearly defined focus. It shows that anti-microbial oils may have potential for inhibiting S. aureus in calcium sufate beads. The results are also relevant, in addition to the medical aspects, in the food sector.

- Thank you!

Minor comment: line 77 and 79 S. aureus in italics.

- Thanks. Corrected! 

Reviewer #2: The study is quite impressive and scientifically elaborated. I advise some minor corrections/clarifications:

- Mention the plants under study in the material and methods section. DONE

- Mention scientific names of the plants in Table 1 DONE

- Include some representative images of treatments displaying significantly higher ZOIs compared to vancomycin 30 µg control. DONE. Added fig1

-Include representative image of PEO seeded beads. DONE Added fig2

- What was the basis of selecting 30 µg of vancomycin as control? 30 �g is the minimum inhibitory concentration for S. aureus species and is used to conduct antibiograms in research and in the clinic to determine susceptibility/resistance. Also, the filter disks come pre-loaded with this amount of vancomycin, and a reference page for determining the susceptibility. 

- Thank you for your compliment!

6. PLOS authors have the option to publish the peer review history of their article (what does this mean?). If published, this will include your full peer review and any attached files.

Do you want your identity to be public for this peer review? For information about this choice, including consent withdrawal, please see our Privacy Policy.

Reviewer #1: Yes: Per Saris

- Thank you! 

Reviewer #2: No

- Thank you!

End of Responses

Once again, thank you for the review process and helpful comments and feedback. We look forward to publishing the manuscript. Please do not hesitate to reach out if we have missed anything, or if any additional feedback arises. We would be more than happy to address any other issues.

---

## [Decision Letter · Decision Letter 1]

14 Feb 2022

PONE-D-21-14096R1Calcium sulfate beads made with antibacterial essential oil-water emulsions exhibit growth inhibition against Staphylococcus aureus in agar pour plates.PLOS ONE

Dear Dr. Matthew C Swearingen,

Thank you for submitting your manuscript to PLOS ONE. After careful consideration, we feel that it has merit but does not fully meet PLOS ONE’s publication criteria as it currently stands. Therefore, we invite you to submit a revised version of the manuscript that addresses the points raised during the review process. The paper is interesting, however a further revision is necessary. Several monor error are presence (S. aureus not in italic line 74: "orthopedic pathogens, including S. aureus (10,14–16)."; ect...)Is not reported the diameter of the plat used (90 mm or 120 mm).Introduction_Line 77-80: this are "result"...and cannot be insert in the introduction.In the results there are repetition of "materials and methods"Please submit your revised manuscript by 31 March 2022. If you will need more time than this to complete your revisions, please reply to this message or contact the journal office at plosone@plos.org. Please include the following items when submitting your revised manuscript:A rebuttal letter that responds to each point raised by the academic editor and reviewer(s). You should upload this letter as a separate file labeled 'Response to Reviewers'.A marked-up copy of your manuscript that highlights changes made to the original version. You should upload this as a separate file labeled 'Revised Manuscript with Track Changes'.An unmarked version of your revised paper without tracked changes. You should upload this as a separate file labeled 'Manuscript'.

We look forward to receiving your revised manuscript.

Kind regards,

Filippo Giarratana

Academic Editor

PLOS ONE

Reviewers' comments:

Reviewer's Responses to Questions

**Comments to the Author**

1. If the authors have adequately addressed your comments raised in a previous round of review and you feel that this manuscript is now acceptable for publication, you may indicate that here to bypass the “Comments to the Author” section, enter your conflict of interest statement in the “Confidential to Editor” section, and submit your "Accept" recommendation.

Reviewer #1: All comments have been addressed

Reviewer #3: (No Response)

2. Is the manuscript technically sound, and do the data support the conclusions?

Reviewer #1: Yes

Reviewer #3: Partly

3. Has the statistical analysis been performed appropriately and rigorously? 

Reviewer #1: Yes

Reviewer #3: No

4. Have the authors made all data underlying the findings in their manuscript fully available?

Reviewer #1: Yes

Reviewer #3: Yes

5. Is the manuscript presented in an intelligible fashion and written in standard English?

Reviewer #1: Yes

Reviewer #3: Yes

6. Review Comments to the Author

Reviewer #1: No further requirements.You have answered the requests in a proper manner. The work presents interesting results.

Reviewer #3: General comments:

Overall, I like the idea on which the work is based. The study tests the possible in vitro antimicrobial activity of several essential oils incorporated into calcium sulphate beads against S. aureus that is among the main causative agent of bone infections in orthopaedic medicine. the authors demonstrated a good command of scientific English. However, there are some limitations. Meanwhile, the results are described on the basis of a not well-defined statistical analysis. Or rather, in the tables this information is reported, however, there is no information about it in the materials and methods. After that, my main concern is that the authors tested multiple essential oils in the same plate. How can they be sure that there has not been an influence between the various oils? I understand that the halos of inhibition are well defined in the plates, however the diffusion of the essential oil in the medium certainly affected the growth.

These are important and relevant methodological limits for the reliability of the results.

Please check the punctuation and italic. I have the impression that a few dots are missing through the text (see lines 65 and 75,77,79 etc.).

Introduction

This paragraph is well written and well explains the background on which the study moves. I appreciated how the aims achieved and the limits of experimentation are briefly mentioned in the final part.

However, I suggest checking, and eventually rephrasing, the lines 79-81. Maybe, at the beginning of line 80, “or” was “of”? I understand what the Authors want to say but is a little bit confusing.

Materials and methods

Line 101 “through preliminary testing”. What are you referring to? Where is this data? Explain better.

Line 102 “ZOIs”: is this the abbreviation of what? Do you mean zone of inhibition? It does not open to me is written by anyone else before. To specify. The same for other abbreviation used such as PPZOI in the results section.

Lines 101-102 “Some PEOs known to have high potency (through preliminary testing) were

102 tested on plates in the absence” Were the other PEOs tested in common plates? If so, how can you be sure the effect of each PEO does not affect that of the other? Please, clarify.

I don't understand what kind of statistical analysis was done. Although there is some clarification in the tables, the type of statistical analysis that was performed is not specified in this section. Please, improve.

Results

Overall, the data relating to the results are presented, however more than one sentence is not about the results but more about the materials and methods. In addition, there are also repetitions of the materials and methods. Results certainly need to be improved.

Lines 135-139: These are not results; furthermore, these are repetitions of materials and methods. See also lines 189-191.

Line 140: It is not clear how these statistically significant differences were calculated. What results were compared with each other? Are there any replicas of the samples? How much they were? In addition to reporting this information in the tables, it is necessary to report it in the text to understand not only the type of analysis used but whether it is actually possible to use this type of analysis for the type of data available. Specify and improve.

Discussions

The discussions are well written, and the data reported in the results are properly argued. Furthermore, both the limitations and the innovations brought about by this study are highlighted.

Figures

The quality of the figures is poor. Improve.

7. PLOS authors have the option to publish the peer review history of their article (what does this mean?). If published, this will include your full peer review and any attached files.

Reviewer #1: No

Reviewer #3: No

---

## [Author Response · Author response to Decision Letter 1]

2 Jun 2022

Dear (new) Editor and Reviewers,

Thank you for the opportunity to revise our manuscript “Calcium sulfate beads made with antibacterial essential oil-water emulsions exhibit growth inhibition against Staphylococcus aureus in agar pour plates.” We appreciate your feedback and the feedback of the reviewers, we are also grateful for you and the reviewers taking the time to conduct a review process for this manuscript. We have considered the helpful comments and addressed them to the best of our ability. As requested in the decision letter email, we are providing responses to the items highlighted in the feedback about the manuscript. 

Update 26Apr2022 - Request from Editor with Response in Red

We've checked your submission and before we can proceed, we need you to address the following issues:

We note that the grant information you provided in the ‘Funding Information’ and ‘Financial Disclosure’ sections do not match.

This work was funded with departmental funds from the Department of Biological Sciences at Florida Gulf Coast University, and for which there is no specific grant award number. Therefore, I am selecting “no specific funding for this work. 

31Mar2022 - Review Feedback and Comments pasted from decision letter email with our responses in red.

• Several monor error are presence (S. aureus not in italic line 74: "orthopedic pathogens, including S. aureus (10,14–16)."; ect...)

o Fixed! Thank you!

• Is not reported the diameter of the plat used (90 mm or 120 mm).

o Fixed line 105

• Introduction_Line 77-80: this are "result"...and cannot be insert in the introduction.

o Fixed

• In the results there are repetition of "materials and methods"

o Fixed lines 138-143 and 175-177

o complete

o complete

o complete

We look forward to receiving your revised manuscript.

Kind regards,

Filippo Giarratana

Academic Editor

PLOS ONE

Reviewers' comments:

Reviewer's Responses to Questions

Comments to the Author

1. If the authors have adequately addressed your comments raised in a previous round of review and you feel that this manuscript is now acceptable for publication, you may indicate that here to bypass the “Comments to the Author” section, enter your conflict of interest statement in the “Confidential to Editor” section, and submit your "Accept" recommendation.

Reviewer #1: All comments have been addressed

Reviewer #3: (No Response)

2. Is the manuscript technically sound, and do the data support the conclusions?

Reviewer #1: Yes

Reviewer #3: Partly

3. Has the statistical analysis been performed appropriately and rigorously?

Reviewer #1: Yes

Reviewer #3: No

4. Have the authors made all data underlying the findings in their manuscript fully available?

Reviewer #1: Yes

Reviewer #3: Yes

5. Is the manuscript presented in an intelligible fashion and written in standard English?

Reviewer #1: Yes

Reviewer #3: Yes

6. Review Comments to the Author

Reviewer #1: No further requirements.You have answered the requests in a proper manner. The work presents interesting results.

• Thank you

Reviewer #3: General comments:

Overall, I like the idea on which the work is based. The study tests the possible in vitro antimicrobial activity of several essential oils incorporated into calcium sulphate beads against S. aureus that is among the main causative agent of bone infections in orthopaedic medicine. the authors demonstrated a good command of scientific English. However, there are some limitations. Meanwhile, the results are described on the basis of a not well-defined statistical analysis. Or rather, in the tables this information is reported, however, there is no information about it in the materials and methods. After that, my main concern is that the authors tested multiple essential oils in the same plate. How can they be sure that there has not been an influence between the various oils? I understand that the halos of inhibition are well defined in the plates, however the diffusion of the essential oil in the medium certainly affected the growth.

These are important and relevant methodological limits for the reliability of the results.

Summary of Responses (followed by individual section responses)

• The student’s T test was used to determine statistical significance and this was previously written in the table 1 title in bold. The specific null hypotheses or otherwise were also described underneath the table. 

• I added a note about using student’s T test in each methods subsection lines 115 and 135.

• Statistics were also previously written throughout the body of the manuscript, for example, lines 144 – 145

• Lines 188-192 address possible synergism/antagonism in multi-PEO plates. Although not within the scope of this study, we thank you for the call out and agree that acknowledging this limitation strengthens the discussion.

Please check the punctuation and italic. I have the impression that a few dots are missing through the text (see lines 65 and 75,77,79 etc.).

• This was noted by the editor, and was completed. Thank you!

Introduction

This paragraph is well written and well explains the background on which the study moves. I appreciated how the aims achieved and the limits of experimentation are briefly mentioned in the final part.

• Thank you!

However, I suggest checking, and eventually rephrasing, the lines 79-81. Maybe, at the beginning of line 80, “or” was “of”? I understand what the Authors want to say but is a little bit confusing.

• Fixed. Thank you for the call-out.

Materials and methods

Line 101 “through preliminary testing”. What are you referring to? Where is this data? Explain better.

• Fixed the wording to not be less vague or misleading.

Line 102 “ZOIs”: is this the abbreviation of what? Do you mean zone of inhibition? It does not open to me is written by anyone else before. To specify. The same for other abbreviation used such as PPZOI in the results section.

• Defined ZOI line 111

• Defined PPZOI 133

Lines 101-102 “Some PEOs known to have high potency (through preliminary testing) were

• Fixed the wording, also addressed above

102 tested on plates in the absence” Were the other PEOs tested in common plates? If so, how can you be sure the effect of each PEO does not affect that of the other? Please, clarify.

• Lines 188-192 address possible synergism/antagonism in multi-PEO plates. Although not within the scope of this study, we thank you for the call out and agree that acknowledging this limitation strengthens the discussion.

I don't understand what kind of statistical analysis was done. Although there is some clarification in the tables, the type of statistical analysis that was performed is not specified in this section. Please, improve.

• I added a note about using student’s T test in each methods subsection lines 115 and 135.

Results

Overall, the data relating to the results are presented, however more than one sentence is not about the results but more about the materials and methods. In addition, there are also repetitions of the materials and methods. Results certainly need to be improved.

• Fixed

Lines 135-139: These are not results; furthermore, these are repetitions of materials and methods. See also lines 189-191.

• Fixed

Line 140: It is not clear how these statistically significant differences were calculated. What results were compared with each other? Are there any replicas of the samples? How much they were? In addition to reporting this information in the tables, it is necessary to report it in the text to understand not only the type of analysis used but whether it is actually possible to use this type of analysis for the type of data available. Specify and improve.

• Again, we used student’s t test and this is stated several times in the manuscript, and we now added lines to both methods subsections explicitly stating this. Table 1 was designed to comprehensively show the ZOI data and how comparisons used for p values. 

o For instance, a T test was used to determine that [for arborvitae] the disk diffusion assay ZOI was significantly higher than a vancomycin control. 

Discussion

The discussions are well written, and the data reported in the results are properly argued. Furthermore, both the limitations and the innovations brought about by this study are highlighted.

Figures

The quality of the figures is poor. Improve.

o This comment is subjective and/or too vague to enact specific change. The images were taken with a high resolution camera, and are at the maximum resolution of 300 dpi. Some oils do not elicit a perfect margin for ZOIs or especially PPZOIs which are 3 dimensional, and can create and impression of blurriness. At this point, I can’t retroactively improve image quality, nor is it acceptable to “doctor” images. 

7. PLOS authors have the option to publish the peer review history of their article (what does this mean?). If published, this will include your full peer review and any attached files.

Do you want your identity to be public for this peer review? For information about this choice, including consent withdrawal, please see our Privacy Policy.

Reviewer #1: No

Reviewer #3: No

Once again, thank you for the review process and helpful comments and feedback. We look forward to publishing the manuscript. Please do not hesitate to reach out if we have missed anything, or if any additional feedback arises. We would be more than happy to address any other issues. 

Most Sincerely,

Matthew C. Swearingen, PhD

p.s. Editor, this work was originally completed at Florida Gulf Coast University in Fort Myers, Florida. I have since left that institution and now work in private industry as a Senior Microbiologist. Hence, I am no longer using the FGCU contact information, but instead my personal information for correspondences.

---

## [Editor Report · Decision Letter 2]

27 Jun 2022

Calcium sulfate beads made with antibacterial essential oil-water emulsions exhibit growth inhibition against Staphylococcus aureus in agar pour plates.

PONE-D-21-14096R2

Dear Dr. Matthew C Swearingen,

We’re pleased to inform you that your manuscript has been judged scientifically suitable for publication and will be formally accepted for publication once it meets all outstanding technical requirements.

Kind regards,

Filippo Giarratana

Academic Editor

PLOS ONE

Additional Editor Comments (optional):

The authors have improved the paper with all the required suggestion/revision.

The paper now can be accepted.

---

## [Editor Report · Acceptance letter]

30 Jun 2022

PONE-D-21-14096R2 

Calcium sulfate beads made with antibacterial essential oil-water emulsions exhibit growth inhibition against *Staphylococcus aureus* in agar pour plates. 

Dear Dr. Swearingen:

I'm pleased to inform you that your manuscript has been deemed suitable for publication in PLOS ONE. Congratulations! Your manuscript is now with our production department. 

Kind regards, 

on behalf of

Dr. Filippo Giarratana 

Academic Editor

PLOS ONE